# Deciphering the Role of ADAMTS6 in the Epithelial–Mesenchymal Transition of Lung Adenocarcinoma Cells

**DOI:** 10.3390/ijms262411850

**Published:** 2025-12-08

**Authors:** Kirill V. Odarenko, Anastasiya M. Matveeva, Grigory A. Stepanov, Marina A. Zenkova, Andrey V. Markov

**Affiliations:** Institute of Chemical Biology and Fundamental Medicine, Siberian Branch of the Russian Academy of Sciences, 630090 Novosibirsk, Russia; k.odarenko@yandex.ru (K.V.O.); anastasiya.maatveeva@gmail.com (A.M.M.); stepanovga@1bio.ru (G.A.S.); marzen@1bio.ru (M.A.Z.)

**Keywords:** ADAMTS, EMT, lung cancer, metastasis, drug resistance

## Abstract

A disintegrin and metalloproteinase with thrombospondin motifs 6 (ADAMTS6) is an extracellular matrix (ECM) protease that promotes the invasion of lung adenocarcinoma (LUAD) cells. Herein, we investigate its role in epithelial-mesenchymal transition (EMT), a process that drives metastasis and drug resistance in LUAD. Re-analysis of microarray and RNA sequencing data from LUAD cells revealed that during EMT, TGF-β1 increased *ADAMTS6* expression, presumably through the SMAD pathway, as SMAD2 loss completely blocked this effect. Moreover, *ADAMTS6* was shown to occupy hub positions within TGF-β1-associated gene networks. Using additional datasets, we found that *ADAMTS6* expression increased under other EMT-inducing conditions, including IL-1β induction and acquired gefitinib resistance, but decreased upon knockdown of Twist1, a master regulator of EMT. Knockout of *ADAMTS6* repressed colony formation, migration, invasion, and doxorubicin resistance but enhanced cell–ECM adhesion in A549 cells. This effect was mediated by EMT inhibition, evidenced by upregulation of E-cadherin and downregulation of N-cadherin, vimentin, and Twist1, and was accompanied by suppressed nuclear translocation of the NF-κB p65 subunit. Re-analysis of transcriptomic data from patient tumors demonstrated that high *ADAMTS6* expression correlated with the expression of EMT markers, further supporting the *ADAMTS6*–EMT link. Moreover, high *ADAMTS6* expression was associated with worse survival prognosis. Overall, *ADAMTS6* promotes EMT in LUAD cells and may be considered a marker of this process, as well as a potential therapeutic target for its inhibition.

## 1. Introduction

Lung cancer was the most common form of cancer worldwide in 2022, accounting for 2.48 million new cases, with about half being lung adenocarcinoma (LUAD) [1]. LUAD often metastasizes to such organs as brain, bone, and liver even in its early stages, which complicates effective treatment [2]. Understanding the molecular mechanisms underlying metastasis offers valuable insights for monitoring and preventing the progression of LUAD.

Epithelial-mesenchymal transition (EMT) is a process in which cells lose apical-basal polarity and cell–cell contacts and gain mesenchymal properties, such as front-rear polarity and increased motility. In cancer, EMT promotes tumorigenesis, metastasis, and drug resistance [3].

Disintegrin and metalloproteinase with thrombospondin motifs 6 (ADAMTS6) is an extracellular protease important for cardiac and muscular development but has recently been shown to participate in cancer progression [4]. ADAMTS6 promotes breast and colon cancer cell invasion [5,6] by inducing EMT through the AKT and NF-κB pathways [6]. Studies in non-tumor cells have shown that ADAMTS6 can activate the latent transforming growth factor beta 1 (TGF-β1) and downstream SMAD2/3 pathway [7,8], whose role in EMT has been demonstrated in numerous studies [9,10,11,12,13].

Although ADAMTS6 affects invasion and drug resistance in LUAD cells [14,15], its relation to EMT is unclear. However, TGF-β1 and tumor necrosis factor alpha (TNF-α) upregulate ADAMTS6, which is crucial for inducing invasion in A549 cells [15]. In patient lung tumors, ADAMTS6 expression positively correlates with the mesenchymal marker vimentin [15]. The related proteases ADAMTS1 and ADAMTS16 activate latent TGF-β1, inducing EMT in LUAD cells [16,17]. These observations raise the possibility that ADAMTS6 may be a marker and/or regulator of EMT in LUAD cells, which deserves further investigation.

This study aimed to investigate the role of ADAMTS6 in EMT to elucidate its potential as a prognostic marker and therapeutic target in LUAD. To this end, *ADAMTS6* expression and its position within the gene network of TGF-β1-induced EMT were evaluated through analysis of previously published microarray data. Publicly available microarray, RNA-sequencing (RNA-seq), and single-cell RNA-sequencing (scRNA-seq) datasets of LUAD cells were used to verify ADAMTS6 as a marker of EMT. Patient data were also analyzed to further validate the association of *ADAMTS6* with classical EMT markers, clinicopathological characteristics, and survival outcomes. Finally, *ADAMTS6* was knocked out in A549 cells using CRISPR/Cas9 methodology to assess its effect on the acquisition of mesenchymal traits by the cells.

## 2. Results

### 2.1. Assessment of the Role of ADAMTS6 in EMT of LUAD Cells

A comprehensive bioinformatic analysis of publicly available transcriptomic data was performed to investigate the potential of *ADAMTS6* as a marker and regulator of EMT in LUAD cells.

First, we performed differential expression analysis comparing TGF-β1-treated H358, A549, and HCC827 human LUAD cells to untreated controls using microarray datasets GSE79235, GSE49644, GSE114761, and GSE123031. H358 cells were treated with two TGF-β1 regimens: a full dose of 4 ng/mL for 2 wk (yielding 1906 differentially expressed genes (DEGs)) and a half-dose of 2 ng/mL for 3 wk (yielding 1032 DEGs). Both regimens shared a core transcriptomic response of 643 DEGs, while 1263 and 389 DEGs were specific to the full dose and half-dose, respectively. For A549 cells, pooled samples from 24-, 48-, and 120 h time points with 5 ng/mL TGF-β1 treatment were compared to pooled control samples, revealing 556 DEGs. HCC827 cells treated with 10 ng/mL TGF-β1 for 72 h exhibited 437 DEGs (Figure 1A). Notably, only 46 DEGs were shared among the three cell lines, highlighting the strong dependence of TGF-β1-induced transcriptomic response on cell context.

Next, we validated the induction of EMT in the analyzed datasets. Gene set enrichment analysis (GSEA) revealed that the EMT hallmark signature was enriched in TGF-β1-treated cells compared to controls across all four datasets (Figure 1B). While the specific set of differentially expressed EMT markers varied across datasets, a consistent pattern emerged: epithelial markers (e.g., *CDH1*, *EPCAM*, and *KRT19*) showed decreased expression, while mesenchymal markers (e.g., *CDH2*, *VIM*, and *FN1*) showed increased expression (Figure 1C). These data demonstrate that TGF-β1 induces EMT in LUAD cells, regardless of the treatment regimen, which is consistent with the original studies [18,19,20].

We then examined how induction of EMT by TGF-β1 affected *ADAMTS6* expression. As shown in Figure 1D, H358 and HCC827 cells showed a similar level of ADAMTS6 upregulation after TGF-β1 stimulation (log_2_FC ≥ 2.7). The same trend was observed in A549 cells (Figure 1D), with *ADAMTS6* expression increasing from 24 to 72 h of TGF-β1 induction (Figure 1E). Although this trend did not reach statistical significance, it was confirmed in verification datasets (see Section 2.3 below). Taken together, *ADAMTS6* upregulation may be considered a marker of TGF-β1-induced EMT in LUAD cells.

To assess the regulatory potential of *ADAMTS6*, we performed a gene network analysis based on the assumption that genes acting as hubs within the EMT network are key regulators of this process [21]. We constructed four gene networks from DEGs obtained from four datasets using STRING database and visualized them in Cytoscape (Appendix A). Network topology was analyzed using CytoHubba plugin, and genes belonging to the top 15% according to at least one of twelve centrality metrics were selected as hub candidates. Intersecting the candidate lists from four networks revealed 50 hub genes common to at least three networks (Figure 2A), of which 23 were upregulated, 23 were downregulated, and 4 showed variable expression across the datasets after TGF-β1 induction (Figure 2B). Importantly, *ADAMTS6* was identified as the hub gene upregulated in three out of four datasets. Most of its centrality metrics were moderate, but DMNC and Clustering Coefficient showed high values, suggesting a stable coordinating role within the networks (Figure 2C).

We hypothesized that upregulated hub genes play a leading role in EMT of LUAD cells. To test this hypothesis, we analyzed Pubmed abstracts using GenCLiP 3.0 text mining platform to identify co-occurrence of hub gene names with terms “LUAD”, “lung cancer”, “EMT”, and “metastasis” in the same sentence. Accordingly, 15 hub genes were found to be strongly associated with “lung cancer”, including 10 linked to “LUAD”, and 18 hub genes were strongly associated with both “metastasis” and “EMT” (Figure 2D). Among the hub genes with established literature links were well-known mesenchymal markers (*CDH2* and *MMP9*), EMT transcription factors (*SNAI2* and *ZEB2*) [22], EMT-inducing cytokines (*IL11* [23], *WNT5B* [24], *CCL20* [25], and *CXCL1* [26]), and genes with established EMT-regulatory functions (*SERPINE1* [27], *THBS1* [28], and *TIMP2* [29]). In contrast, *ADAMTS6*, as well as *CGB8*, *PCOLCE2*, *PTPRK*, and *COL7A1*, were rarely mentioned together with the studied terms in the literature, indicating that they are understudied in the context of EMT (Figure 2D). Taken together, the text mining results confirmed that the network analysis identified not only established EMT regulators but also new players. Particularly, *ADAMTS6* had connectivity within EMT networks comparable to that of established EMT regulators, suggesting a strong regulatory function.

The subsequent study is devoted to verifying two facts established in this section, namely that (1) *ADAMTS6* upregulation is a marker of EMT in LUAD cells, and (2) *ADAMTS6* is a positive regulator of EMT in LUAD cells.

### 2.2. Verification of ADAMTS6 as a Marker of EMT in Lung Epithelial Cells

We employed additional datasets to verify the role of *ADAMTS6* as a marker of TGF-β1-induced EMT of lung cells. As shown in Figure 3A, treatment with TGF-β1 at a concentration of 2 ng/mL increased *ADAMTS6* expression in H1975 LUAD cells after 24–48 h (log_2_FC ≥ 1.3) and in SK-MES-1 squamous cell carcinoma cells after 6–48 h (log_2_FC ≥ 1.2). However, the greatest upregulation occurred in normal small airway epithelial cells upon chronic exposure to TGF-β1 at 10 ng/mL for 15 d (log_2_FC = 6.1). Next, we assessed the contribution of epithelial-like and mesenchymal-like cell types to *ADAMTS6* upregulation by analyzing the GSE147405 scRNA-seq dataset, which included A549 cells collected at 0–7 d of TGF-β1 induction and at 0–3 d after TGF-β1 withdrawal. Seven distinct cell clusters were identified: clusters 1, 2, 3, and 4 were present initially (time point 0 h), while clusters 0, 5, and 6 emerged following TGF-β1 treatment (Figure 3B). EMT status of these clusters was evaluated by single-sample gene set enrichment analysis (ssGSEA): the epithelial gene signature was enriched in clusters 1, 2, and 3, whereas the mesenchymal gene signature and hallmark EMT signature were enriched in clusters 0, 4, 5, and 6 (Figure 3C). Accordingly, clusters 1, 2, and 3 were classified as epithelial-like, while clusters 0, 4, 5, and 6 were classified as mesenchymal-like, consistent with the expression of classical EMT markers (Appendix A). *ADAMTS6* showed higher expression in mesenchymal-like cells compared to epithelial-like cells (Figure 3D), which was also confirmed by pseudobulk analysis (Figure 3E). Taken together, TGF-β1 increased *ADAMTS6* expression in a wide range of lung cells, with upregulation associated with a mesenchymal phenotype, supporting the role of *ADAMTS6* as an EMT marker.

### 2.3. Upstream Regulators of ADAMTS6 Expression During EMT Induced by TGF-β1 and Other Factors

In the next step, we studied the mechanism of *ADAMTS6* regulation during EMT of LUAD cells, starting with the previously discussed model of TGF-β1-stimulated A549 cells. As shown in Figure 4A, chronic TGF-β1 treatment (2 ng/mL, 60 days) caused a much higher induction of *ADAMTS6* expression than short-term induction (5–10 ng/mL, 48 h), which is consistent with the results on lung cell lines (Figure 3A). TGF-β1 induces EMT via the SMAD2 pathway [30], which is accompanied by low production of acetyl-CoA through fatty acid metabolism [31]. In A549 cells, a loss-of-function mutation of SMAD2 blocked TGF-β1-induced *ADAMTS6* upregulation (log_2_FC = −2.6), whereas restoration of acetyl-CoA levels by addition of sodium acetate had no significant effect (Figure 4A). Overall, the effect of TGF-β1 on *ADAMTS6* expression was mediated by SMAD2, independent of metabolic status, and tended to increase over time.

Various tumor microenvironmental factors, not only TGF-β1, can induce EMT, and this process involves many proteins with both inducing and repressing roles [3]. Therefore, we decided to examine *ADAMTS6* expression in broader contexts by analyzing datasets in which (i) EMT was induced by interleukin 1 beta (IL-1β), TNF-α, resistance to EGFR tyrosine kinase inhibitors (EGFR-TKIs), protein kinase C (PKC) activation, or RAVER1 knockdown; or (ii) EMT was inhibited by Twist1 knockdown or YAP1 knockout. Chronic IL-1β induction for 15–21 d increased *ADAMTS6* expression in A549 cells by log_2_FC > 2 (Figure 4B), while short-term TNF-α induction for 24 h had no effect (*p* > 0.05; Appendix A). Resistance to the EGFR-TKI gefitinib upregulated *ADAMTS6* in HCC4006 cells (log_2_FC = 2.5). We examined whether EGF regulates *ADAMTS6*, but neither its addition to H1975 cells nor silencing of its receptor significantly affected *ADAMTS6* expression (log_2_FC = 0.7). We then assessed the influence of MEK on *ADAMTS6* expression, as MEK overactivation is a mechanism underlying EGFR/TKI resistance [32]. The OT regimen (EGFR-TKI osimertinib + MEK inhibitor trametinib, 2 wk) caused significant downregulation (log_2_FC = −2.4) in PC9 cells, suggesting an inductive role of MEK on *ADAMTS6*. Other identified positive regulators of *ADAMTS6* were PKC and the EMT maser regulator Twist1: PKC activation by phorbol 12-myristate 13-acetate (PMA) enhanced *ADAMTS6* expression (log_2_FC = 1.1), whereas Twist1 knockdown abolished it (log_2_FC = −3.6). In turn, the EMT repressor RAVER1 and YAP1, a downstream effector of the Hippo pathway, were identified as negative regulators: their suppression increased *ADAMTS6* levels in LUAD cells, with log_2_FC of 1.7 and 1.2, respectively (Figure 4B). Notably, the OT regimen reduced the effect of YAP1 knockout (log_2_FC = −1.6). Taken together, *ADAMTS6* expression is upregulated by a number of factors (Figure 4C), suggesting that its role as an EMT marker extends beyond TGF-β1 signaling.

### 2.4. The Development of a CRISP/Cas9 System Targeting ADAMTS6

We employed CRISP/Cas9-induced knockout to verify the role of *ADAMTS6* as a positive regulator of EMT, as suggested by in silico network analysis (Figure 2A,C). The *ADAMTS6* gene consists of 31 exons and has multiple splice variants, with the main isoform containing 25 exons encoding a protein of approximately 125.3 kDa. To disrupt the *ADAMTS6* open reading frame (ORF), exon 4 was selected as the target region for CRISPR/Cas9-mediated editing due to its size (169 bp) and its position at the start of the transcript. The knockout strategy involved inducing two double-strand breaks (DSBs) simultaneously, one within intron 3 and another within exon 4 (Figure 5A).

Target protospacer sequences identified via Benchling analysis demonstrated good efficiency/specificity ratios (Figure 5A). Constructs expressing CRISPR/Cas9 components were generated based on pX458 vector following a standard protocol [33]. A549 cells were transfected according to the proposed strategy. Subsequently, GFP-expressing cell fractions were sorted to obtain single-cell clones. Initial screening involved PCR analysis using primers flanking the expected deletion region. Several clones showed loss of genomic sequence between protospacers (“4-1” + “e-b”) (Figure 5B). The size of deletions varied among clones and did not always match expectations; for example, clones 202 and 204 exhibited larger deletions. A second round of PCR, using primers closely flanking the exon, demonstrated the presence of a second *ADAMTS6* allele in all clones except one. The target region structure of clone 202 was further analyzed via Sanger sequencing of PCR products (Figure 5C). The results revealed two mutations: a 1047 bp deletion covering parts of exon 4 and introns 3 and 4, and an inversion between DSB sites that disrupts the exon structure (Appendix A). Both mutations are predicted to result in the loss of functional ADAMTS6 protein.

*ADAMTS6* knockout was confirmed via RT-qPCR using primers located downstream of the established mutational sites. As shown in Figure 5D, clone 202 exhibited a 2.2-fold reduction in *ADAMTS6* expression compared to the parental A549 line. Lachat et al. previously demonstrated that sequential incubation with 4 ng/mL TGF-β1 for 3 d and 20 ng/mL TNF-α for 2 d significantly induces *ADAMTS6* expression in A549 cells [15]. Using this protocol, we observed a 2.6-fold increase in *ADAMTS6* expression in control A549 cells. In contrast, TGF-β1/TNF-α treatment did not affect *ADAMTS6* expression in clone 202, which remained lower than in untreated A549 cells (Figure 5D). Overall, CRISP/Cas9 disrupted the *ADAMTS6* gene structure and abrogated its stable expression in clone 202 derived from A549 cells, presumably through nonsense-mediated decay [34]. In the next sections, we used clone 202, referred to as AD6-KO cells, to evaluate the effect of *ADAMTS6* knockout on EMT.

### 2.5. Functional Assessment of ADAMTS6 Knockout in LUAD Cells

We then investigated which cellular functions were affected by *ADAMTS6* knockout. Microscopic examination showed no difference in morphology between AD6-KO and parental A549 cells (Appendix A). In normal culture, AD6-KO and A549 cells exhibited the same proliferative capacity, as assessed by MTT (Appendix A). However, at low density, AD6-KO cells formed 2.9 times smaller colonies than those of A549 cells, which may indicate reduced metastatic potential (Figure 6A). We further examined whether *ADAMTS6* knockout affects cellular migration and invasion, key features of metastasis and EMT. In the wound healing and transwell assays, migration of AD6-KO cells was slower compared to A549 cells at 24 h and 48 h (Figure 6B,C). In the Matrigel-coated transwell assay, invasion was observed starting at 72 h, and thereafter AD6-KO cells invaded more slowly than A549 cells, with a statistically significant difference of 29.6% observed at 83 h (Figure 6D).

Cell–ECM adhesion underlies cell migration. However, mechanistic studies show that only moderate adhesion promotes cell migration, while fully mature adhesion favors attachment to the extracellular matrix (ECM), resulting in slower migration [35]. We performed a cell-matrix adhesion assay to evaluate whether *ADAMTS6* knockout affects cell adhesion. Compared with A549 cells, the adhesion of AD6-KO cells to plastic, collagen, and Matrigel was 46.8%, 31.3%, and 48.6% higher, respectively (Figure 6E). Therefore, *ADAMTS6* knockout suppresses cell migration and invasion, probably due to reinforced cell–ECM adhesion.

### 2.6. Mechanism of EMT Mediated by ADAMTS6 Expression

We next examined whether *ADAMTS6* affects the expression of EMT-associated markers. RT-qPCR indicated that the expression of the epithelial marker *CDH1* increased by 3.8-fold, while the expression of the mesenchymal markers *CDH2*, *VIM*, and *TWIST1* decreased by 0.7-, 0.7-, and 0.6-fold, respectively, in AD6-KO cells compared to A549 cells (Figure 7A). Expression of cytokeratins *KRT18* and *KRT19* did not differ between cell lines (Appendix A). In immunofluorescence (IF) staining, AD6-KO cells had lower level of vimentin, encoded by *VIM*, than A549 cells (Figure 7B).

Since *ADAMTS6* induces EMT in colon cancer cells via the NF-κB pathway [6] and acts as an inducer of the TGF-β1/SMAD2/3 pathway [7,8], we investigated whether these pathways were affected by *ADAMTS6* knockout. IF showed a higher nuclear-to-cytoplasm intensity ratio of the NF-κB p65 subunit in A549 cells than in AD6-KO cells, indicating that *ADAMTS6* knockout reduced p65 nuclear translocation (Figure 7C). In contrast, nuclear translocation of SMAD2/3 was not affected (Appendix A). In summary, *ADAMTS6* knockout suppressed EMT and converted cells to a more epithelial-like phenotype, likely through inhibition of the NF-κB pathway.

In LUAD cells, EMT induces resistance to the topoisomerase II inhibitor doxorubicin [36,37]. MTT assay showed that doxorubicin was more toxic to AD6-KO cells than to A549 cells, with IC_50_ values of 1.45 μM and 2.4 μM, respectively (Figure 6D). Thus, inhibition of EMT by *ADAMTS6* knockout resulted in a more doxorubicin-sensitive phenotype.

Taken together, our results show that *ADAMTS6* knockout inhibits EMT (Figure 6 and Figure 7), supporting the role of *ADAMTS6* as a regulatory gene in EMT of LUAD cells.

### 2.7. Validating Associations of ADAMTS6 with EMT in Patient Cohort

Next, we validated the association of *ADAMTS6* with EMT by analyzing expression data from human LUAD tumors in the TCGA-LUAD, GSE31210, and GSE72094 datasets. *ADAMTS6* expression was positively correlated with mesenchymal markers (*CDH2*, *VIM*, and *FN1*) and EMT-related transcription factors (*SNAI1*, *SNAI2*, *ZEB1*, and *ZEB2*), but negatively correlated with epithelial markers (*CDH1*, *EPCAM*, *OCLN,* and *KRT19*) (Figure 8A). In each dataset, we selected 20 samples with the highest and 20 samples with the lowest *ADAMTS6* expression (Appendix A), and then performed GSEA to analyze pathways enriched in *ADAMTS6*-high versus *ADAMTS6*-low groups. The hallmark EMT signature was enriched in all three datasets (Figure 8B). Among the enriched Kyoto Encyclopedia of Genes and Genomes (KEGG) and Gene Ontology (GO) terms, seven were common to three datasets (Figure 8C), while others were dataset-specific (Appendix A). These common terms were related to metallopeptidase activity, focal adhesion, ECM, and cell–ECM interaction. These results support the functional data for *ADAMTS6* obtained in our knockout model, linking *ADAMTS6* expression to gene programs regulating EMT and cell adhesion.

Finally, we analyzed the association between *ADAMTS6* expression and clinicopathological characteristics in LUAD patients. In TCGA-LUAD dataset, *ADAMTS6* expression was higher in tumors than in normal lung tissues (Figure 8D), which was also confirmed using UALCAN and OncoDB databases (Appendix A). *ADAMTS6* expression did not differ between patients of different age, gender, pathological stage, or TNM stage (Appendix A). We used GEPIA2 and KMplotter platforms to classify patients from TCGA-LUAD (GEPIA2) and microarray datasets (KMplotter) into two groups based on median *ADAMTS6* expression and compared their survival. In TCGA-LUAD, the *ADAMTS6*-high group showed shorter disease-free survival (DFS) than the *ADAMTS6*-low group (Figure 8E). A similar trend was observed for overall survival (OS), but it was not statistically significant (*p* = 0.095). In the pooled microarray cohort, the *ADAMTS6*-high group demonstrated worse progression-free survival (PFS) and OS (Figure 8F). These results suggest that *ADAMTS6* overexpression is an unfavorable prognostic factor in LUAD patients.

## 3. Discussion

Metastasis is a life-threatening complication of cancer, particularly LUAD, and investigating its molecular mechanisms is crucial for developing improved prognostic and therapeutic methods [2]. Recent studies show that the metastatic potential of cancer cells is increased after their EMT [9,10,11,12,13]. This process is accompanied by increased secretion of ECM proteases, which allow cancer cells to migrate and invade the surrounding tissues [38,39]. The most studied ECM proteases, metalloproteinases (MMPs), have been shown to induce EMT, creating a positive feedback loop [40]. Unfortunately, clinical trials of MMP inhibitors for cancer treatment have revealed intolerable side effects, which can be explained by the crucial role of MMPs in normal tissue homeostasis [41]. The search for tumor-specific ECM proteases may help improve cancer prognosis and identify new therapeutic targets.

Compared to MMPs, the ADAMTS family of ECM proteases has been significantly less studied in relation to cancer. In LUAD, overexpression of ADAMTS1 [42], ADAMTS8 [43], or ADAMTS18 [44] is associated with better patient survival, while overexpression of ADAMTS4 [45], ADAMTS5 [46], or ADAMTS16 [16] is associated with worse survival. An attractive idea for treating LUAD is to selectively target only those ADAMTS proteases that promote cancer progression. Such therapies are already being developed for the treatment of osteoarthritis, with inhibitors of ADAMTS4 and ADAMTS5 currently in phase III clinical trials [47]. To this end, it is important to better characterize which ADAMTS proteases contribute to LUAD progression, as well as the molecular mechanisms underlying their effects. Recent studies represent an important step forward in this process, demonstrating that ADAMTS1 [17], ADAMTS4 [45], and ADAMTS16 [16] induces EMT of LUAD cells. While ADAMTS1 and ADAMTS16 mediate their effects by promoting TGF-β1 maturation from its latent form [16,17], ADAMTS4 activates the MAPK pathway [45]. Both ADAMTS1 and ADAMTS16, promote lung cancer metastasis in animal models: ADAMTS1 accelerates spontaneous pulmonary metastasis of Lewis lung carcinoma in C57BL/6 mice, while ADAMTS16 enhances pulmonary metastasis of A549 cells following tail vein injection in nude mice [16,48]. However, contradictory results were reported by Lee et al., who showed EMT induction in A549 cells after *ADAMTS1* knockout [42]. Interestingly, *ADAMTS5* knockdown suppressed migration and invasion of A549 cells while downregulating E-cadherin and upregulating vimentin, respectively [46].

Our study found that increased *ADAMTS6* expression could serve as a marker of EMT in LUAD cells. TGF-β1 induced *ADAMTS6* expression in LUAD cells (Figure 1D,E, Figure 3A and Figure 4A), which was characteristic of cells that acquired a mesenchymal-like phenotype (Figure 3D,E). The upregulation of *ADAMTS6* by TGF-β1 was dependent on SMAD2 (Figure 4A) and was significantly enhanced by TNF-α (Figure 5D) [15]. It has been previously shown that the TGF-β1/SMAD2/3 and TNF-α/NF-κB pathways crosstalk in LUAD cells undergoing EMT [49,50]. Both TGF-β and TNF-α increase ADAMTS6 expression in normal epithelial cells [51,52]. In our microarray analysis, single agent TNF-α did not affect *ADAMTS6* expression in A549 cells (Appendix A).

*ADAMTS6* upregulation was not limited to TGF-β1 induction but was also observed in LUAD cells after EMT was induced by other conditions (Figure 4B). IL-1β significantly increased *ADAMTS6* expression, while the EGF/EGFR axis had little effect. In comparison, both IL-1 and EGF upregulate *ADAMTS6* in keratinocytes [51]. Conversely, ADAMTS6 inhibits EGFR activation without altering its expression in breast cancer cells [53]. Resistance to the EGFR-TKI gefitinib strongly elevates *ADAMTS6* expression in LUAD cells. Since EGFR-TKI resistance is mediated by hyperactivation of the MEK/ERK pathway [32], we showed that dual EGFR/MEK inhibition reduced *ADAMTS6* expression, suggesting that MEK acts as an upstream regulator of ADAMTS6. Supporting this, nicotinamide N-methyltransferase (NNMT) decreases phosphatase PP2A methylation, which induces EMT via MEK activation in breast cancer cells [54] and promotes migration while increasing *ADAMTS6* expression in renal cancer cells [55]. RAVER1, which regulates alternative splicing, acts as an EMT suppressor [56]. RAVER1 knockdown increased *ADAMTS6* expression in LUAD cells, which may be related to the upregulation of TGF-β1 and its receptors TβRI and TβRII upon loss of RAVER1 [56]. We showed that *ADAMTS6* expression is elevated by PMA-induced activation of PKC. This effect may also be mediated through the TGF-β1 pathway, as PKC has been shown to induce TGF-β1 secretion [57] and increase the expression of the transcription factors Snail and Twist, which act downstream of TGF-β1 [58]. Accordingly, Twist1 knockdown abrogated *ADAMTS6* expression in LUAD cells. Contradictory results were also obtained. Although YAP1 mediates EMT induced by mechanical stress [59] or EGFR/MEK inhibition [60], *YAP1* knockout increased *ADAMTS6* expression in LUAD cells. Since ADAMTS6 activates YAP in TC28a/2 chondrocytes [7], it can be speculated that ADAMTS6 and YAP regulate each other in a negative feedback loop.

Our study also showed that the *ADAMTS6* gene plays a regulatory role in EMT of LUAD cells. This effect was initially demonstrated through in silico gene network analysis (Figure 2), a method that has previously been successful in identifying EMT regulators such as CRTC2 [61], MDK [62], and POLR3G [63]. The connectivity of *ADAMTS6* within EMT gene networks was comparable to known regulatory genes of EMT, such as *SNAI2*, and *ZEB2* [22], demonstrating its strong regulatory potential. We verified the results obtained in silico by knocking out *ADAMTS6* in A549 cells using CRISP/Cas9. Migration and invasion, key features of EMT, were significantly reduced by *ADAMTS6* knockout (Figure 6B–D). We suggest that this effect could be related to the elevation in cell–ECM adhesion (Figure 6E), as it was previously shown that fully mature adhesive contacts attach cells to ECM rather than promoting movement [35]. Consistent with this, ADAMTS6 inhibits focal adhesions in normal epithelial cells by cleaving syndecan-4 [8]. Interestingly, transcriptomic analysis of LUAD patient tumors showed a correlation between the activation of focal adhesion genes and high *ADAMTS6* expression (Figure 8C). This can be explained by a compensatory response of cells to restore cell adhesion. By cleaving syndecan-4, ADAMTS6 reduces activation of focal adhesion kinase (FAK) [8]. To compensate, cells may upregulate Pyk2, a structurally similar kinase, or FAK-related non-kinase (FRNK), both of which can partially substitute for FAK functions [64,65]. The role of the ADAMTS6/syndecan-4/FAK axis in EMT requires further study.

Since EMT induces resistance of LUAD cells to doxorubicin [36], we also showed that *ADAMTS6* knockout increased the sensitivity of A549 cells to doxorubicin (Figure 7D). In contrast, ADAMTS6 knockdown increases the expression of AGR2, which mediates acquired resistance to gefitinib and suppresses ADAMTS6 expression, thus forming a negative feedback loop [14]. Interestingly, AGR2 inhibits EMT in LUAD cells: it enhances the expression of E-cadherin, suppresses the expression of vimentin, N-cadherin, ZEB1 and Slug, reduces cell invasion, and enhances cell–ECM adhesion [66]. These effects contrast with the *ADAMTS6* activity demonstrated in our study (Figure 6 and Figure 7). These observations raise the possibility that the ADAMTS6/AGR2 axis exerts opposing effects on gefitinib resistance and EMT, which requires further investigation.

We demonstrated that the *ADAMTS6* knockout increased the expression of the epithelial marker *CDH1* (E-cadherin), but decreased the expression of the mesenchymal markers *CDH2* (N-cadherin), *VIM* (vimentin), and *TWIST1* (Twist1) in A549 cells (Figure 7A,B). These findings were supported by the correlation between *ADAMTS6* expression and marker gene expression obtained from transcriptomic data analysis in LUAD patients (Figure 8A,B). Co-expression of ADAMTS6 and vimentin has been previously shown by immunohistochemistry (IHC) method in lung tumors from patients [15]. In colon cancer cells, *ADAMTS6* knockdown increased E-cadherin expression and decreased N-cadherin and vimentin, while *ADAMTS6* overexpression had an opposite effect [6]. In esophageal squamous cell carcinoma, IHC has shown co-expression of ADAMTS6 and Twist1 in patient tumors [67].

Since *ADAMTS6* knockout inhibited characteristic features of EMT, such as migration, invasion, doxorubicin resistance, and expression of EMT markers, we suggest that the *ADAMTS6* gene acts as a positive regulator of EMT in LUAD cells. A similar pro-EMT role of ADAMTS6 has been demonstrated in colon cancer [6]. In contrast, *ADAMTS6* overexpression inhibits EMT in epithelial ovarian cancer [68]. Contradictory results have been obtained in breast cancer studies, with ADAMTS6 demonstrating both pro-EMT [5] or anti-EMT [53] activities.

Our study also revealed a possible molecular mechanism behind the pro-EMT effect of *ADAMTS6* in LUAD. *ADAMTS6* knockout inhibited nuclear translocation of the NF-κB p65 subunit in A549 cells (Figure 7C). Consistent with this, overexpression of *ADAMTS6* enhances activation of p65 through phosphorylation in colon cancer cells [6]. Although ADAMTS6 has previously been shown to activate latent TGF-β1 and induce the downstream SMAD2/3 pathway [7,8], we did not observe changes in SMAD2/3 nuclear translocation upon ADAMTS6 knockout in A549 cells (Appendix A). These results suggest that ADAMTS6 supports EMT of LUAD cells via the NF-κB, but not the SMAD2/3 pathway.

By analyzing clinicopathological parameters, we found that *ADAMTS6* expression did not vary in LUAD tumors from patients of different ages, genders, tumor stages, and TNM stages (Appendix A). However, *ADAMTS6* expression was elevated in LUAD tumors compared to normal lung tissues (Figure 8D). Interestingly, this finding contradicts previous studies that reported no difference between normal and tumor tissues [69,70]. To validate our results, we cross-checked the data using the UALCAN and OncoDb databases [71,72] (Appendix A). Additionally, we observed that high *ADAMTS6* expression was associated with reduced OS, DFS, and PFS in LUAD patients (Figure 8E,F). Therefore, like ADAMTS4 [45], ADAMTS5 [46], and ADAMTS16 [16], ADAMTS6 may serve as a negative prognostic marker for LUAD.

This study has limitations. Although *ADAMTS6* knockout reduced both migration/invasion and cell adhesion, further experiments are needed to establish a causal relationship between these effects (Figure 6B–E). To more fully assess the role of *ADAMTS6* in EMT, the results obtained with the knockout model (Figure 5, Figure 6 and Figure 7) could be complimented with overexpression experiments. Furthermore, it would be valuable to evaluate the effect of *ADAMTS6* on LUAD metastasis in mouse models. These limitations may represent promising avenues for future research.

## 4. Materials and Methods

### 4.1. Chemicals and Reagents

Recombinant human TGF-β1 (CYT-716) and recombinant human TNF-α (PSG250-50) were obtained from ProSpec-Tany TechnoGene Ltd. (Ness-Ziona, Israel) and Sci-Store (Skolkovo, Russia), respectively. The following reagents were used for gene editing using CRISPR/Cas9: pSpCas9(BB)-2A-GFP vector (Addgene, #48138, Watertown, MA, USA), *BstV2I* restriction endonuclease (SibEnzyme, Novosibirsk, Russia), T4 DNA ligase (Thermo Fisher Scientific, Waltham, MA, USA), Plasmid-mini kit (Biolabmix Ltd., Novosibirsk, Russia), HiPure Plasmid EF Maxi kit (Magen Biotech, Guangzhou, China), and Lipofectamine 3000 (Thermo Fisher Scientific, Waltham, MA, USA). PCR and Sanger sequencing reagents included: Fast lysis buffer, BioMaster HS-Taq PCR-Color mix, D-cells kit, DR kit, M-MuLV-RH revertase, RT buffer mix, BioMaster SYBR Blue reagent kit (all from Biolabmix Ltd., Novosibirsk, Russia), TRIzol Reagent (Ambion, Austin, TX, USA), BigDye™ Terminator v3.1 Cycle Sequencing Kit (Thermo Fisher Scientific, Waltham, MA, USA). IF reagents included: rabbit anti-vimentin (ab92547), rabbit anti-SMAD2/3 (ab202445), AlexaFluor 488-conjugated goat anti-IgG (ab150077) antibodies (all from Abcam, Waltham, MA, USA), rabbit anti-p65 (A2547, ABclonal, Wuhan, China), DAPI, and Fluoromount-G (both from Thermo Fisher Scientific, Rockford, IL, USA).

### 4.2. Data Acquisition

Microarray data from TGF-β1-induced H358 cells (GSE79235, GSE49644), A549 cells (GSE114761), HCC827 cells (GSE123031), and patient LUAD tumors (GSE31210 and GSE72094), along with scRNA-seq data from TGF-β1-treated A549 cells (GSE147405) and several verification datasets (Appendix A), were retrieved from Gene Expression Omnibus (GEO) database (https://www.ncbi.nlm.nih.gov/geo/) with accession dates of 22 September 2022 (cell data), 23 August 2025 (patient data), 24 February 2025 (scRNA-seq), and 4 October 2025 (verification data). The Cancer Genome Atlas Lung Adenocarcinoma (TCGA-LUAD) RNA-Seq dataset was sourced from Genomic Data Commons (GDC) data portal (https://portal.gdc.cancer.gov/, accessed on 23 August 2025).

### 4.3. Microarray Analysis

Expression data were normalized using log normalization. Differential expression analysis of the cell culture datasets GSE79235, GSE49644, GSE114761, and GSE123031 was performed using GEO2R tool, applying a threshold of |log_2_FC| ≥ 1.5 and a Benjamini-Hochberg corrected *p*-value < 0.05. In the verification datasets, changes in *ADAMTS6* expression during EMT in LUAD, lung squamous carcinoma, and small airway epithelium cells were analyzed using GEO2R, with significant regulation defined as |log_2_FC| ≥ 1 and a Benjamini-Hochberg corrected *p*-value < 0.05. The design of these comparisons is presented in Appendix A. For clinical datasets GSE31210 and GSE72094, differential expression analysis was conducted by selecting the top 20 and bottom 20 samples based on *ADAMTS6* expression and comparing these groups using limma package (v3.62.2). GSEA was performed on DEGs to assess the enrichment of the hallmark EMT gene set, as well as gene sets from Kyoto Encyclopedia of Genes and Genomes (KEGG) and Gene Ontology molecular function (GO-MF).

### 4.4. Gene Set Enrichment Analysis

All gene sets were obtained from Molecular Signatures Database (MSigDB) using msigdbr package (v25.1.1). GSEA was performed on DEGs ranked by log_2_FC using clusterProfiler package (v4.14.6) with the following parameters: gene set sizes ranging from 25 to 500, multiple testing correction using the Benjamini-Hochberg method, and a *p*-value cutoff of 0.05.

### 4.5. Gene Network Analysis

Gene networks were constructed using DEGs from GSE79235, GSE49644, GSE114761, and GSE123031 with STRING plugin (v2.2.0) in Cytoscape platform (v3.10.3), applying a confidence score > 0.7 and no additional interactions as cutoff criteria. Gene centrality was assessed using twelve topological criteria from cytoHubba plugin (v0.1).

### 4.6. Text Mining

The co-occurrence of hub gene names with the terms “lung adenocarcinoma OR LUAD”, “epithelial-mesenchymal transition OR EMT”, “lung cancer”, and “metastasis” within the same sentence in the MEDLINE database was evaluated using GenCLiP 3.0 (http://cismu.net/genclip3/analysis.php, accessed on 6 March 2025). A gene was considered linked to a term if more than five articles mentioning both were found.

### 4.7. scRNA-Seq Analysis

scRNA-seq analysis was performed using Seurat package (v5.3.0). High-quality cells were included in the analysis if they had more 200 but less than 5000 detected genes, over 1000 UMI counts, and less than 5% mitochondrial gene content. Raw data were log-normalized, and expression values of all genes were scaled. Dimensionality reduction was conducted using principal component (PC) analysis on the top 2000 variable genes. Cells were clustered using the top 50 PCs and a resolution of 1 via FindNeighbors and FindClusters functions. To determine whether cell clusters were epithelial-like or mesenchymal-like, ssGSEA was performed using irGSEA package (v3.3.2) with default settings, assessing enrichment of the hallmark EMT set from MSigDB, along with epithelial and mesenchymal gene sets identified by Cook et al. [73]. Clusters were then merged into epithelial-like and mesenchymal-like groups, and pseudobulk differential expression analysis was performed using edgeR algorithm from DElegate package (v1.2.1), applying a threshold of |log_2_FC| ≥ 0.5 and a false discovery rate (FDR)-adjusted *p*-value < 0.05.

### 4.8. sgRNA Design and Cloning Strategy

sgRNAs targeting intron 3 (“4-1”) and exon 4 (“e-b”) of the *ADAMTS6* gene were designed using Benchling (https://www.benchling.com, accessed on 25 October 2022) (Appendix A). Potential off-target effects were assessed using the Cas-OFFinder tool [74]. sgRNAs with high on-target and specificity scores were selected, and a minimum of three mismatches was required for any predicted off-target site. Top and bottom strands of protospacers were annealed and cloned into CRISPR/Cas9-expressing vector pSpCas9(BB)-2A-GFP (pX458) as described previously [33]. The vector was pre-digested with *BstV2I* restriction endonuclease (SibEnzyme), and ligation was performed using T4 DNA ligase (Thermo Fisher Scientific). Electrocompetent *E. coli* TOP10 cells were transformed using MicroPulser (Bio-Rad, Hercules, CA, USA). Plasmids were isolated using Plasmid-mini (Biolabmix Ltd.) and HiPure Plasmid EF Maxi (Magen Biotech) kits. DNA concentration and quality were assessed using a Nano-500 spectrophotometer (Allsheng Instruments Co., Ltd., Hangzhou, China). Insertions into plasmids were verified by Sanger sequencing.

### 4.9. Cell Lines and Transfection

Human A549 LUAD cells were obtained from Russian Cell Culture Collection (St. Petersburg, Russia) and maintained in DMEM (Sigma-Aldrich Inc., St. Louis, MO, USA) supplemented with 10% fetal bovine serum (FBS; Dia-M, Moscow, Russia), 100 U/mL penicillin, 100 μg/mL streptomycin, and 0.25 μg/mL amphotericin B (Central Drug House, New Delhi, India) at 37 °C in 5% CO_2_. A549 cells (1 × 10^6^) were plated in 6-well plates and, after 24 h, transfected with 2 μg of expression vectors containing the “4-1” and “e-b” inserts mixed at a 1:1 ratio using Lipofectamine 3000 (Thermo Fisher Scientific) according to the manufacturer’s protocol. Cells transfected with pX458 plasmid lacking the spacer sgRNA region served as the transfection control. After 48 h, cells were sorted by FACS (S3e Cell Sorter, Bio-Rad, Hercules, CA, USA), and 72 h later, single GFP-positive cells were plated in 96-well plates at a density of 0.5 cells per well. After 10–14 d of growth in full DMEM/F12, single-cell clones were divided into two 96-well plates for maintenance and mutation verification, respectively.

### 4.10. Verification of CRISP/Cas9-Generated Mutation

DNA was extracted from clones using 100 µL of Fast lysis buffer (Biolabmix Ltd.) supplemented with Proteinase K, following the manufacturer’s instructions. Routine PCR amplification was performed using BioMaster HS-Taq PCR-Color mix (Biolabmix Ltd.) and primers listed in Appendix A. Mutations were identified by analyzing PCR products separated by electrophoresis on a 1.5% agarose gel and stained with ethidium bromide.

Clones harboring mutations were selected for Sanger sequencing using 4F/4R primers (Appendix A). Samples preparation involved isolating genomic DNA using D-cells kit (Biolabmix Ltd.), followed by PCR amplification and purification of the resulting amplicons using DR kit (Biolabmix Ltd.).

### 4.11. Sanger Sequencing

Sequencing was performed using BigDye™ Terminator v3.1 Cycle Sequencing Kit (Thermo Fisher Scientific). Reaction products were purified on Sephadex G-50 columns (Cytiva, Uppsala, Sweden) and analyzed using an ABI 3130XL Genetic Analyzer (Thermo Fisher Scientific Inc., Waltham, MA, USA) at SB RAS Genomics Core Facility.

### 4.12. Quantitative Real-Time Polymerase Chain Reaction (qRT-PCR)

To study ADAMTS6 expression, cells were seeded in 12-well plates for 24 h, then cultured in DMEM with 5% FBS and treated with 4 ng/mL TGF-β (ProSpec-Tany Technogene Ltd.) for 3 d followed by 20 ng/mL TNF-α (Sci-Store) for 2 d. To evaluate EMT marker expression, cells were cultured in DMEM with 10% FBS for 24 h. After the end of incubation periods, total RNA was isolated from cells using TRIzol Reagent (Ambion) according to the manufacturer’s protocol. First strand cDNA was synthesized using 4 μg total RNA, 0.1 μM random hexaprimer, 5000 U/mL M-MuLV-RH revertase, and RT buffer mix (Biolabmix Ltd.). RT-qPCR was performed using gene specific primers (Appendix A) and BioMaster SYBR Blue reagent kit (Biolabmix Ltd.). Relative expression was determined by normalizing to the housekeeping gene *GAPDH* using the 2^−ΔΔCT^ method.

### 4.13. Evaluation of Cell Morphology

Cells (2500) were plated in 96-well plates for 24 h. Next, images were captured using EVOS XL Core microscope and 200 cells from five random fields of view were quantified for shape (aspect ratio, AR) and size using ImageJ.

### 4.14. Cell Viability Analysis

Cells (10^4^) were cultured in 96-well plates in DMEM supplemented with 10% FBS for 24 and 48 h. Then, MTT dye was added at a concentration of 0.5 mg/mL and incubated for 3 h. After incubation, the medium was removed, formazan crystals were dissolved in DMSO, and the optical density was measured at 570 nm using Multiscan RC plate reader (Thermo LabSystems, Helsinki, Finland).

### 4.15. Colony Formation

Cells were plated at low density (300 cells per well) in 6-well plates and incubated for 10 d, with medium refreshed every 5 d with 10% FBS-supplemented DMEM. Colonies were fixed with 4% formaldehyde and stained with 0.1% (*w*/*v*) crystal violet dye. Images were captured using iBright 1500 Imaging System (Thermo Fisher, Waltham, MA, USA), and colony area was measured using ImageJ.

### 4.16. Wound Healing Assay

Cells (1.3 × 10^5^) were seeded in 24-well plates and incubated for 24 h to reach a confluent monolayer. Wounds were created on the monolayers using 10 μL pipette tip, and detached cells were removed by washing with PBS. The cells were then cultured in serum-free DMEM for 24 and 48 h, and their migration toward the wounds was captured using EVOS XL Core microscope with a built-in camera (Thermo Fisher Scientific, Waltham, MA, USA). Wound closure was quantified by measuring the wound area at each time point using ImageJ (v2.14.0) and normalizing it to the wound area at 0 h.

### 4.17. Transwell Assays

For migration assays, 2 × 10^4^ cells suspended in serum-free DMEM were placed in the upper chamber of CIM-plate, while DMEM with 10% FBS was added to the lower chamber. The invasion assay followed a similar protocol except that the bottom of the upper chamber was coated with a 1:5 dilution of Matrigel (BD Biosciences, Bedford, MA, USA) in DMEM for 1 h before adding the cells. Cell migration and invasion to the lower chamber were monitored using xCELLigence RTCA DP system (ACEA Biosciences Inc., San Diego, CA, USA).

### 4.18. Adhesion Assay

Cells (5 × 10^4^) were seeded in 96-well plates coated with Matrigel, rat tail collagen (Cell Applications Inc., San Diego, CA, USA), or left uncoated. After 1 h, non-adherent cells were removed by washing three times with PBS, and adherent cells were quantified by MTT assay.

### 4.19. Immunofluorescence

Cells (2 × 10^5^) were seeded on glass coverslips for 24 h and then fixed with 4% formaldehyde. Rabbit antibodies against vimentin (1:200, ab92547), p65 (1:100, A2547), and SMAD2/3 (1:100, ab202445) were added to coverslips in permeabilization buffer (PBS supplemented with 0.1% Triton X-100 and 5 mg/mL BSA) for 1 h. Then, AlexaFluor 488-conjugated goat anti-IgG antibody (1:500, ab150077) was added in PBS supplemented with 5 mg/mL BSA for 1 h. Nuclei were stained by incubation with DAPI (Thermo Fisher Scientific) at 1 μg/mL for 10 min. After each staining step, cells were washed three times with BSA-supplemented PBS. Coverslips were mounted in Fluoromount-G (Thermo Fisher Scientific) and imaged using a LSM710 confocal microscope (Zeiss, Oberkochen, Germany) at ×200 and ×650 magnifications.

### 4.20. Bulk RNA-Seq Analysis

Expression data from the TCGA-LUAD dataset were normalized using variance stabilizing transformation (VST) method. Subsequently, the top 20 and bottom 20 samples were selected based on *ADAMTS6* expression levels. Raw counts were then extracted for these selected samples, and differential expression analysis comparing high and low *ADAMTS6* expression groups was performed using DESeq2 package (v1.46.0). Subsequently, GSEA was performed on DEGs to assess the enrichment of the hallmark EMT gene set, as well as gene sets from Kyoto Encyclopedia of Genes and Genomes (KEGG) and Gene Ontology molecular function (GO-MF).

### 4.21. Survival Analysis

The impact of *ADAMTS6* on LUAD patient survival was assessed using GEPIA2 (http://gepia2.cancer-pku.cn/, accessed on 25 June 2025) for RNA-Seq TCGA data and KMplotter (https://kmplot.com/analysis/, accessed on 25 June 2025) for 17 TCGA and GEO microarray datasets (JetSet best probe set). Patients were divided based on median *ADAMTS6* expression, and overall, progression-free, and disease-free survival were analyzed using Kaplan–Meier curves and the log-rank test, using *p*-value < 0.05 as a significance threshold.

### 4.22. Statistical Analysis and Visualization

Data analysis and visualization was conducted using R v4.4.2 and RStudio v.2024.12.1. Pairwise comparisons were conducted using a two-tailed Student’s *t*-test from the stats package (v4.4.2). Spearman correlation analysis was performed, with results visualized using Hmisc (v5.2.3) and corrplot (v0.95) packages. Additional visualization tools included ggplot2 (v4.0.0), pheatmap (v1.0.13), circlize (v0.4.16), and ggvenn (v0.1.10) packages.

## 5. Conclusions

In this study, *ADAMTS6* emerged as a marker and positive regulator of EMT in LUAD. It was consistently upregulated under diverse EMT-inducing conditions. The central position of *ADAMTS6* within EMT gene networks suggested its regulatory potential, which was further validated in vitro. Knockout of *ADAMTS6* reduced colony formation, migration, invasion, and mesenchymal marker expression while increasing epithelial E-cadherin expression and enhancing doxorubicin sensitivity in A549 cells. This was accompanied by decreased nuclear translocation of p65, suggesting the involvement of the NF-κB pathway in the regulation of EMT by *ADAMTS6*. Heightened *ADAMTS6* expression correlated with poorer survival prognosis in LUAD patients. Together, these findings highlight the importance of *ADAMTS6* in LUAD progression and suggest its potential as a prognostic marker and therapeutic target.

## Figures and Tables

**Figure 1 ijms-26-11850-f001:**
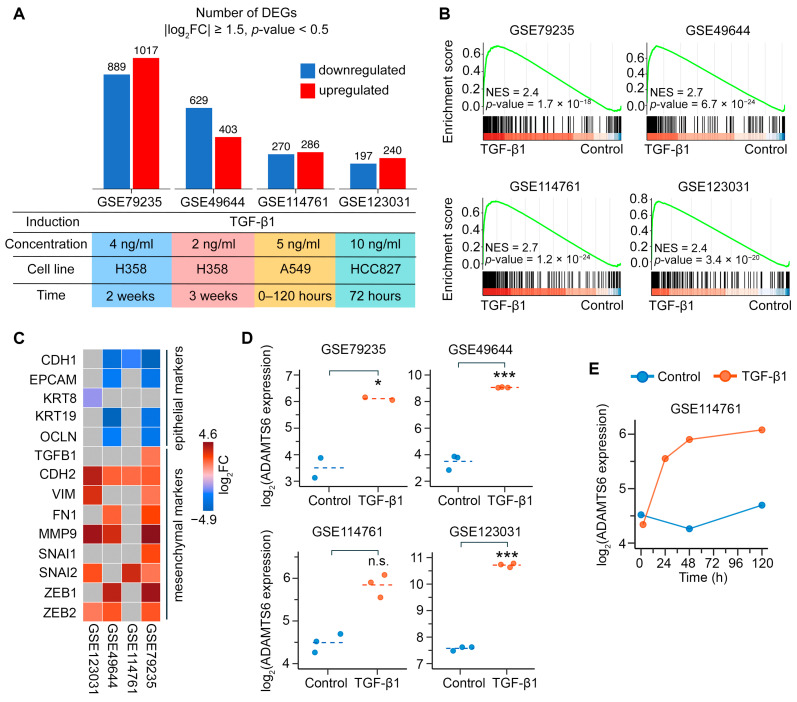
Dissecting the regulatory role of ADAMTS6 in TGF-β1-induced EMT of LUAD cells. (**A**) Differential expression analysis of TGF-β1-treated versus untreated human LUAD cells (top). The table reflects the experimental conditions used in each dataset (bottom). (**B**) Activation of EMT hallmark gene signature in the microarray datasets, as determined by GSEA. (**C**) Differential expression (|log_2_(fold change (FC))| ≥ 1.5) of EMT markers between TGF-β1-induced and control cells in microarray datasets. Genes with non-significant changes (*p*-value ≥ 0.05) are indicated in gray. (**D**) Expression of *ADAMTS6* in control and TGF-β1-induced cells. * and *** indicate *p*-values of <0.05 and <0.001, respectively, based on differential expression analysis. (**E**) Time-dependent changes in *ADAMTS6* expression in A549 cells following TGF-β1 treatment.

**Figure 2 ijms-26-11850-f002:**
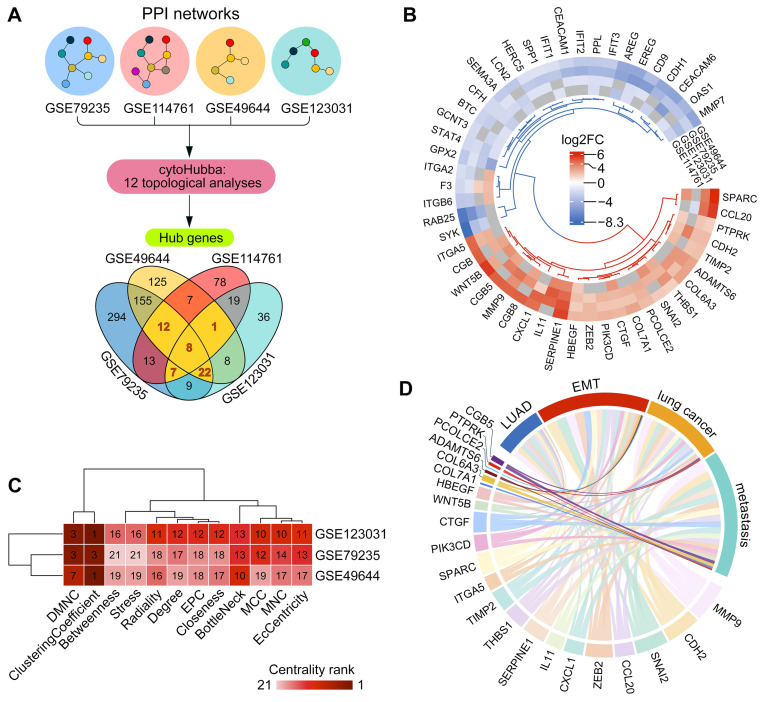
Gene network analysis. (**A**) Analysis workflow. Four gene networks were constructed from DEGs identified in each dataset by comparing TGF-β1-treated and control cells using STRING plugin in Cytoscape, and visualized in Appendix A. Each network was analyzed with CytoHubba plugin, and genes ranked in the top 15% by at least one of twelve centrality criteria were classified as hub genes. The intersection of hub genes from four networks yielded 50 common hub genes. (**B**) Differential expression (|log_2_FC| ≥ 1.5) of 50 common hub genes between TGF-β1-induced and control cells in microarray datasets. Genes with non-significant changes (*p*-value ≥ 0.05) are indicated in gray. (**C**) Ranking of *ADAMTS6* among 23 upregulated hub genes according to 12 centrality measures computed by cytoHubba in gene networks. (**D**) Text mining summary. Associations of common hub genes with EMT, metastasis, LUAD, and lung cancer were extracted from PubMed Central. Line widths reflect the number of sentences in which a hub gene and term co-occur.

**Figure 3 ijms-26-11850-f003:**
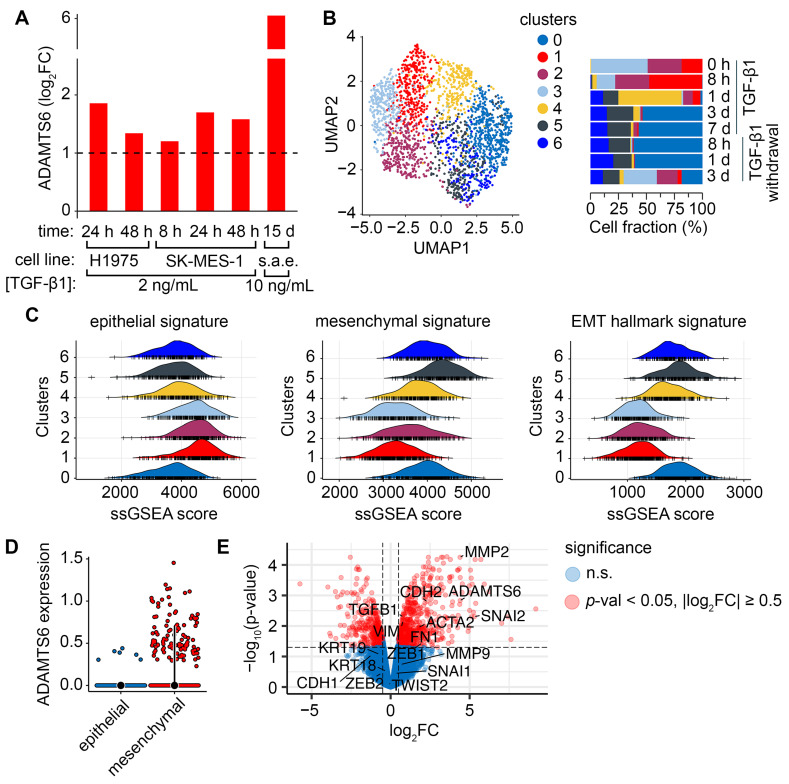
*ADAMTS6* as a marker of EMT. (**A**) Changes in *ADAMTS6* expression in H1975 LUAD, SK-MES-1, and small airway epithelial (s.a.e.) cells following treatment with TGF-β1 at specified concentrations and time points, compared to untreated controls. The dashed line indicates significant upregulation (|log_2_FC| ≥ 1). (**B**) UMAP embeddings of A549 cells (left) and a graph showing the relative representation of clusters under each experimental condition (right). (**C**) Enrichment of epithelial, mesenchymal, and hallmark EMT signatures in cell clusters evaluated by ssGSEA. (**D**) Normalized expression of *ADAMTS6* in epithelial and mesenchymal cells. (**E**) Volcano plot showing genes differentially expressed in mesenchymal cells, with *ADAMTS6* and classical EMT markers highlighted. Information on datasets, group comparisons, and treatment schemes is summarized in Appendix A.

**Figure 4 ijms-26-11850-f004:**
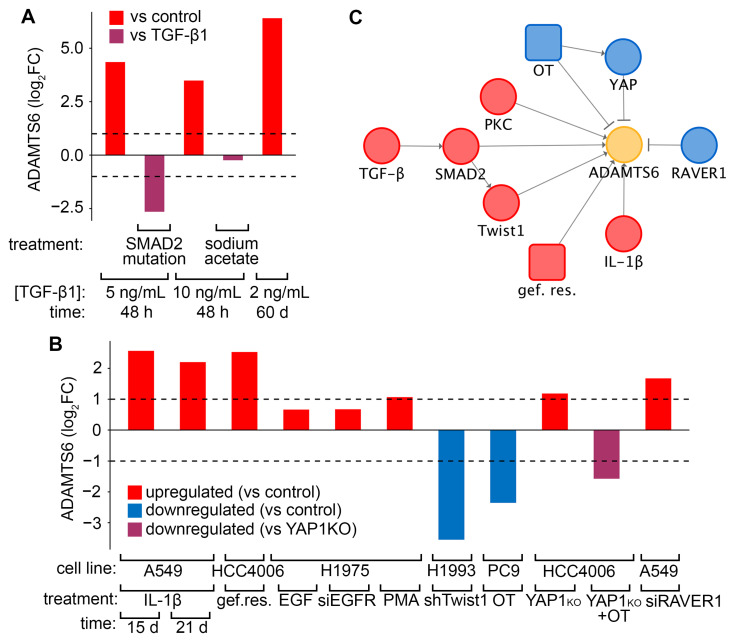
Regulation of *ADAMTS6* expression during EMT. (**A**) Changes in *ADAMTS6* expression in A549 cells following treatment with TGF-β1 and sodium acetate at specified concentrations and time points, and after SMAD2 mutation. Comparisons were made to untreated controls (red) or TGF-β1-treated cells (dark pink). (**B**) Changes in *ADAMTS6* expression in specified LUAD cells upon EMT induction by various stimuli. Red and blue indicate up- and downregulation compared to corresponding controls (untreated, DMSO-treated, siNC), while dark pink shows downregulation relative to YAP1 knockout. (**C**) Network summarizing the regulation of *ADAMTS6* by EMT-mediating genes and factors; red and blue indicate positive and negative regulators of expression, respectively. Gef. res., gefitinib resistance; OT, osimertinib and trametinib treatment; siEGFR and siRAVER1, knockdown of EGFR and RAVER1 by siRNA; shTwist1, Twist1 knockdown by shRNA; YAP1KO, CRISP/Cas9-induced YAP1 knockout. Information on datasets, group comparisons, and treatment schemes is summarized in Appendix A. Dashed lines in (**A**,**B**) indicate significant expression changes (|log_2_FC| ≥ 1).

**Figure 5 ijms-26-11850-f005:**
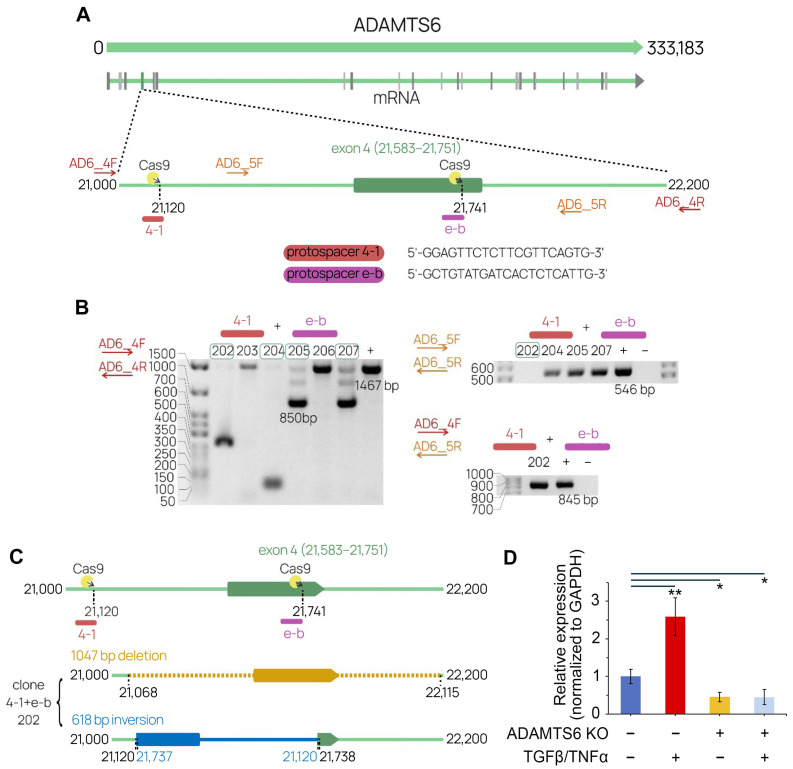
Generation of *ADAMTS6* knockout lines based on A549 human cells. (**A**) Editing strategy involving the usage of two CRISPR-induced DSBs with corresponding sequences of protospacers. (**B**) Clone selection based on PCR analysis of target genome region using various flanking primers. Products were analyzed in 1.5% agarose gel. “+” denotes the control A549 cells used as a positive PCR control. (**C**) CRISPR-induced mutations were determined through Sanger sequencing (see Appendix A). (**D**) Relative expression of *ADAMTS6* in control and AD6-KO cells incubated under regimens: (i) TGF-β (4 ng/mL) for 3 d + TNF-α (20 ng/mL) for 2 d, (ii) without induction (5 d). Assessment by RT-qPCR with normalization to the housekeeping gene *GAPDH*. All experiments were conducted with at least three biological replicates. Data are presented as mean ± SD. Statistical significance was determined by a two-tailed Student’s *t*-test: * *p* < 0.05, ** *p* < 0.01.

**Figure 6 ijms-26-11850-f006:**
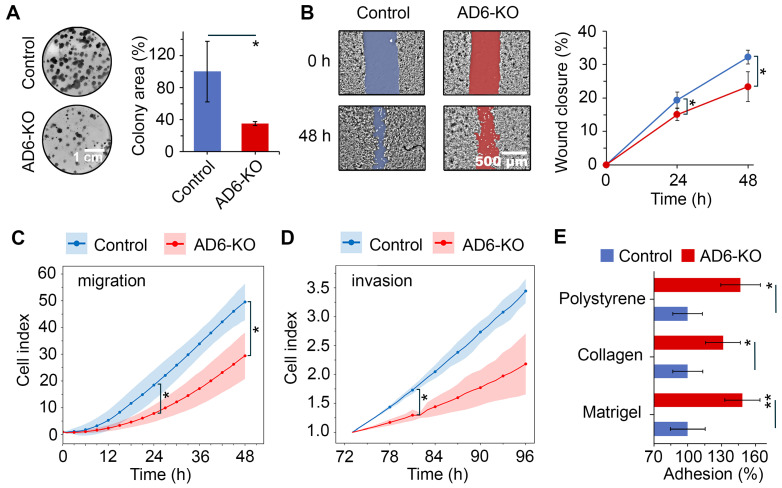
Effect of *ADAMTS6* knockout on metastatic potential. Control, parental A549 cells; AD6-KO, A549 cells with *ADAMTS6* gene knockout. (**A**) Colony formation after 10 d of incubation. (**B**) Cell migration assessed by wound healing assay. (**C**) Cell migration assessed in real-time using xCelligence system, with the cell index normalized to the 1 h time point. (**D**) Cell invasion through Matrigel coating assessed in real-time using xCelligence system, with the cell index normalized to the 72 h time point. (**E**) Cell adhesion to polystyrene, collagen, and Matrigel. After 1 h of incubation on the substrate, non-adherent cells were removed by washing, and the remaining cells were assessed using MTT assay. All experiments were conducted with at least three biological replicates. Data are presented as mean ± SD. Statistical significance was determined by a two-tailed Student’s *t*-test: * *p* < 0.05, ** *p* < 0.01.

**Figure 7 ijms-26-11850-f007:**
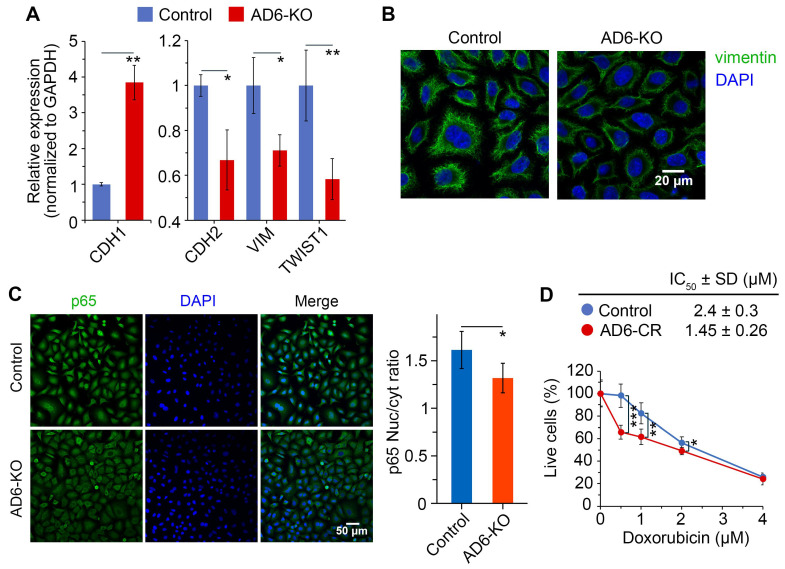
Effect of *ADAMTS6* knockout on EMT. Control, parental A549 cells; AD6-KO, A549 cells with *ADAMTS6* gene knockout. (**A**) Relative expression of the epithelial marker *CDH1* and the mesenchymal markers *CDH2*, *VIM*, and *TWIST1*, assessed using RT-qPCR and normalized to *GAPDH* expression. (**B**) IF staining of vimentin (green) and DAPI staining of nuclei (blue). (**C**) IF staining of NF-κB p65 (green) and DAPI staining of nuclei (blue). p65 nuclear translocation was assessed by the nuclear to cytoplasm (Nuc/cyt) fluorescence ratio. (**D**) Cell viability after 72 h of incubation with specified concentrations of doxorubicin assessed by MTT assay. IC_50_ data are presented in the table. All experiments were conducted with at least three biological replicates. Data are presented as mean ± SD. Statistical significance was determined by a two-tailed Student’s *t*-test: * *p* < 0.05, ** *p* < 0.01, *** *p* < 0.001.

**Figure 8 ijms-26-11850-f008:**
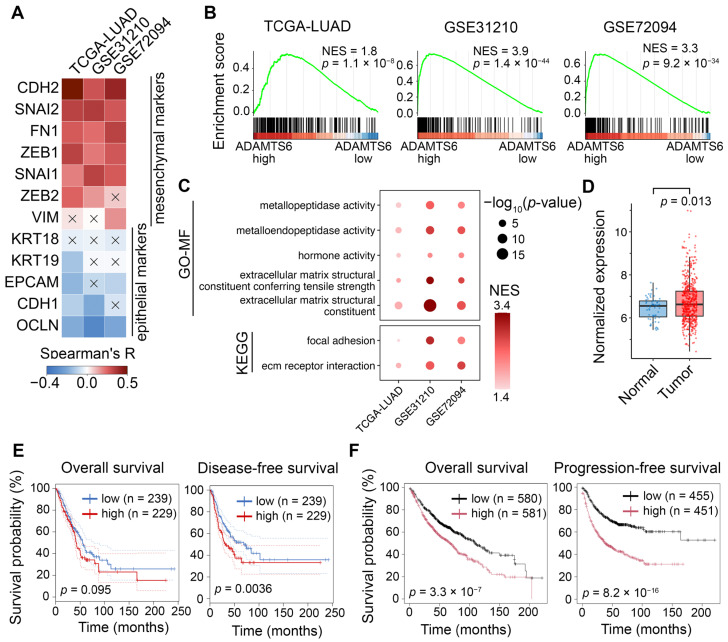
Correlations of *ADAMTS6* expression with EMT and clinicopathological features in LUAD patient cohorts (TCGA-LUAD, GSE31210, and GSE72094). (**A**) Heatmap showing the correlation between expression of ADAMTS6 and EMT markers in patient tumors. Non-significant correlations (*p*-value ≥ 0.05) are indicated by a cross. (**B**) Activation of the EMT hallmark gene signature in tumors with high *ADAMTS6* expression, as determined by GSEA. (**C**) KEGG and GO-MF terms enriched in three clinical datasets when comparing tumors with high versus low *ADAMTS6* expression. (**D**) *ADAMTS6* expression in normal lung tissues and tumors from TCGA-LUAD dataset. Statistical significance was determined by a two-tailed Student’s *t*-test. (**E**,**F**) Kaplan–Meier survival analysis of ADAMTS6 in LUAD patients stratified by median expression, performed using GEPIA2 (**E**) and KMplotter (**F**).

## Data Availability

Data is contained within the article or Appendix A.

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
