# Peer review of "Deciphering the Role of ADAMTS6 in the Epithelial–Mesenchymal Transition of Lung Adenocarcinoma Cells"

_ijms, 2025, doi:10.3390/ijms262411850_

Round 1
Reviewer 1 Report
Comments and Suggestions for Authors
The study addresses an important topic: the mechanisms that enable malignant cells in the most common form of cancer, lung cancer, to form metastases. The article presents a good set of diverse methods, which are described in sufficient detail. The results are well illustrated, and the paper is neatly formatted. I have only a few minor comments:
- Line 158. Figure 2A shows the workflow for the analysis of four gene networks reconstructed from four transcriptomic datasets (GSE79235, GSE49644, GSE114761, and GSE123031), for which DEGs were identified. However, Figure 2C presents the ranking of ADAMTS6 among 23 upregulated hub genes according to 12 centrality measures computed by cytoHubba in three gene networks only (GSE79235, GSE49644, and GSE123031). Could you please clarify in the manuscript why the gene network based on GSE114761 is not represented?
- Line 289, Figure 4: I recommend increasing the contrast of the labels “4-1” and “e-b” in Figure 4, as they are currently barely visible.
- The figures are not placed immediately after the paragraphs in which they are mentioned, but this is likely due to their large size and will presumably be adjusted by the technical editors. Comments on the Quality of English LanguageThere are a few minor errors in the English language. For example:
Line 839: “Patient” should be “Patients” (probably also in line 836).
Line 857: “suggests” should be replaced with “suggest”.
Author Response
Dear Reviewer #1,
Thank you for taking the time to read and thoroughly analyze our article. We greatly appreciate your positive reception of our research, as well as your invaluable feedback. Addressing your insightful comments and suggestions during the revision has significantly strengthened our work.
We have revised the manuscript accordingly and provide our point-by-point responses below.
- Figure 2A shows the workflow for the analysis of four gene networks reconstructed from four transcriptomic datasets (GSE79235, GSE49644, GSE114761, and GSE123031), for which DEGs were identified. However, Figure 2C presents the ranking of ADAMTS6 among 23 upregulated hub genes according to 12 centrality measures computed by cytoHubba in three gene networks only (GSE79235, GSE49644, and GSE123031). Could you please clarify in the manuscript why the gene network based on GSE114761 is not represented?
Authors: Dear Reviewer #1, we agree that this point was not clearly stated in the original manuscript. ADAMTS6 was not ranked in GSE114761 (A549 cells) because we constructed networks based on DEGs, and ADAMTS6 expression only showed an increasing trend, but not differential expression, in this particular dataset. We have more clearly emphasized this point in the revised manuscript (see p. 4, lines 106–110, 120–121). We note that upon subsequent verification, ADAMTS6 upregulation in A549 was confirmed in other datasets.
- Figure 4: I recommend increasing the contrast of the labels “4-1” and “e-b” in Figure 4, as they are currently barely visible.
Authors: Corrected (see Fig. 5, p. 10).
- The figures are not placed immediately after the paragraphs in which they are mentioned, but this is likely due to their large size and will presumably be adjusted by the technical editors.
Authors: Dear Reviewer #1, Thank you for pointing out this flaw in the manuscript. We have corrected it by splitting the original Figure 3 into new Figures 3 and 4 (see pp. 7, 8) and the original Figure 5 into new Figures 6 and 7 (see pp. 12, 13). In addition, the original Figure 5A has been moved to new Figure 5 (see Fig. 5D, p. 10), and the original Figures 5B and 5F have been moved to Supplementary Figure S3 (see supplementary materials, p. 2). By making these restructurings, we ensure that the figures are closely aligned with the text descriptions for ease of reading.
- There are a few minor errors in the English language. For example: Line 839: “Patient” should be “Patients” (probably also in line 836). Line 857: “suggests” should be replaced with “suggest”.
Authors: Corrected (see line 713 on p. 21, line 734 on p. 22).
We hope that this version of the manuscript will be acceptable for publication.
Thank you very much!
Sincerely,
On behalf of all authors,
Dr. Andrey Markov
Akademgorodok, Russia

Reviewer 2 Report
Comments and Suggestions for Authors
Thank you for the opportunity to review your manuscript. The research question is relevant and the integration of public datasets with functional assays is commendable. However, after careful evaluation, I have several major concerns regarding the scientific rigor, experimental design, interpretation of the data, and clarity of presentation. These issues are substantial and fundamentally limit the strength of the conclusions. I outline my comments below.
1. Mechanistic conclusions are speculative and lack experimental support. Several proposed mechanisms—such as syndecan-4 cleavage, modulation of ECM stiffness, activation of mechanotransduction via YAP/TAZ, or release of latent TGF-β—are only mentioned in the Discussion and are based exclusively on enrichment analyses or literature references. No experiments were performed to validate any of these pathways. As a result, the mechanistic part of the manuscript remains speculative and cannot support the conclusions drawn.
2. The experimental design is incomplete. All functional conclusions rely on one CRISPR knockout model with no independent clones, no rescue experiments, and no overexpression studies. Without such controls, it is difficult to exclude clonal selection, off-target effects, or cell line drift. Current standards in molecular cell biology typically require at least one complementary rescue or gain-of-function approach.
3. Logical inconsistencies in the interpretation of results. The manuscript contains several transitions that are difficult to follow. For example, AD6-KO cells show increased adhesion, yet this is immediately extrapolated to the conclusion that ADAMTS6 decreases adhesion to promote migration. The connection between phenotype and conclusion is not clearly explained, leading to confusion about the underlying logic.
4. Several figures—particularly Figures 3 and 5—are extremely challenging to interpret. Multiple unrelated analyses are combined into single composite figures, panel labels do not follow a clear order, and many plots lack quantitative scales or annotations. Some numerical values cited in the text (e.g., fold changes) cannot be derived from the presented figures. This significantly hinders reproducibility and transparency.
5. The manuscript requires clearer organization and improved presentation. The introduction is overly long and contains background information not directly relevant to ADAMTS6. The Results section would benefit from a more structured narrative that guides the reader through the findings. Figures would need simplification and reorganization to enhance readability.
Author Response
Dear Reviewer #2,
We are profoundly grateful for your thorough analysis of our manuscript and for identifying several aspects that required further development. Your valuable comments and suggestions have been instrumental in significantly strengthening the manuscript and improving its clarity for the reader.
All revisions made in response to your comments and those of the first reviewer have been highlighted in yellow within the text.
Please find our point-by-point responses to your feedback below.
- Mechanistic conclusions are speculative and lack experimental support. Several proposed mechanisms—such as syndecan-4 cleavage, modulation of ECM stiffness, activation of mechanotransduction via YAP/TAZ, or release of latent TGF-β—are only mentioned in the Discussion and are based exclusively on enrichment analyses or literature references. No experiments were performed to validate any of these pathways. As a result, the mechanistic part of the manuscript remains speculative and cannot support the conclusions drawn.
Authors: Dear Reviewer #2, Thank you for pointing out this omission in our methodology. Previous studies have shown that ADAMTS6 activates the TGF-β1/SMAD2/3 pathway in normal epithelial cells and induces EMT via the NF-κB pathway in colon cancer cells (we included this information on p. 2, lines 44–48). We performed immunofluorescence staining and showed that ADAMTS6 knockout inhibited the nuclear translocation of the p65 subunit of NF-κB, but not SMAD2/3 proteins (see p. 13, lines 341–346, Fig. 7; supplementary material p. 2, Fig. S3). We have removed the previously described mechanisms based on enrichment analysis and literature references from the Discussion section due to their speculative nature. Instead, we have focused on the in vitro results and emphasized that, based on these, ADAMTS6 likely mediates EMT via the NF-κB but not SMAD2/3 pathway (see p. 17, lines 508–516).
- The experimental design is incomplete. All functional conclusions rely on one CRISPR knockout model with no independent clones, no rescue experiments, and no overexpression studies. Without such controls, it is difficult to exclude clonal selection, off-target effects, or cell line drift. Current standards in molecular cell biology typically require at least one complementary rescue or gain-of-function approach.
Authors: Dear Reviewer #2, we sincerely thank you for raising this crucial point regarding the proper design of experimental work. We agree that additional experiments utilizing rescue or gain-of-function approaches would significantly strengthen our study. When planning the pipeline for the current project, we relied on previously published works in Int. J. Mol. Sci., where researchers also used the CRISPR/Cas9 system to knockout genes of interest and subsequently worked with a single clone, omitting rescue and overexpression experiments (e.g., [1,2]). Your comment highlights the need for a more thorough literature analysis during the initial stage of experimental design. Unfortunately, we are currently quite limited in our ability to perform the experiments you proposed. However, acknowledging the importance of your remark, we have added a dedicated paragraph to the manuscript describing the study's limitations. In this paragraph, we clearly state the necessity for further, more detailed verification steps (such as ADAMTS6 overexpression) in future work (please see p. 18, lines 529-531).
Regarding the off-target effects of our knockout approach: as mentioned previously, guide RNA design was performed using the Benchling tool which allows for on-target activity assessment as well as specificity analysis [3–5]. Both selected guides generated off-target sites with no fewer than three mismatches, which is typically the threshold for maintaining Cas9 activity [6–8]. Two predicted off-targets with 3 mismatches were located in intronic and intergenic regions; this is acceptable as not only mutations are unlikely they would not affect the cell culture state or behavior in functional studies.
We also independently verified the predictions using Cas-OFFinder [9] which is extensively used in CRISPR research and has proven to be a reliable prognostic tool [10–12]. Cas-OFFinder provided a more in-depth analysis. While no off-target sites with 0-1 mismatches were identified, two sites with two mismatches were predicted. However, the PAM-proximal localization of these mismatches suggested negligible sgRNA activity at these loci as such positioning is poorly tolerated by CRISPR/Cas9. Multiple sites with three mismatches were predicted; upon closer inspection, all were located in intronic or intergenic regions. Thus, the off-target potential of our sgRNAs is low and unlikely to confound subsequent analysis. Additional information about Cas-OFFinder usage has been added to Section 4.7 in Materials and methods (please see p. 19, lines 595-597).
- Logical inconsistencies in the interpretation of results. The manuscript contains several transitions that are difficult to follow. For example, AD6-KO cells show increased adhesion, yet this is immediately extrapolated to the conclusion that ADAMTS6 decreases adhesion to promote migration. The connection between phenotype and conclusion is not clearly explained, leading to confusion about the underlying logic.
Authors: Dear Reviewer #2, We agree that the logical transitions from the knockout phenotype to ADAMTS6 functionality were difficult to follow in the original manuscript. For this reason, we have reduced the number of such transitions. In the revised manuscript, we have focused on the effects of ADAMTS6 knockout (see pp. 11–13, lines 288–299, 312–327, 339–351; pp. 16–17, lines 466–471, 480–481, 490–493, 509–514). Only one transition was retained in the Discussion section, and it is based on the assumption that if ADAMTS6 knockout inhibits EMT, then it logically follows that ADAMTS6 plays a positive, stimulatory role in EMT (see p. 17, lines 501–504). This transition is also supported by a previously published study of colon cancer cells.
- Several figures—particularly Figures 3 and 5—are extremely challenging to interpret. Multiple unrelated analyses are combined into single composite figures, panel labels do not follow a clear order, and many plots lack quantitative scales or annotations. Some numerical values cited in the text (e.g., fold changes) cannot be derived from the presented figures. This significantly hinders reproducibility and transparency.
Authors: Dear Reviewer #2, Thank you for pointing out that our figures were difficult to understand. We have split the original Figure 3 into new Figures 3 and 4 (see pp. 7, 8) and the original Figure 5 into new Figures 6 and 7 (see pp. 12, 13). In addition, the original Figure 5A has been moved to new Figure 5 (see Fig. 5D, p. 10), and the original Figures 5B and 5F have been moved to Supplementary Figure S3 (see supplementary materials, p. 2). We have also increased the font size in many graphs and checked the presence of annotations and scales. To improve the connection between numerical values in the text and figures, we have replaced fold changes to log2(fold change) in the description of bioinformatic results (see p. 4, line 106; p. 6, lines 160–162; pp. 8–9, lines 198, 223, 225, 227, 230, 233–237) and corrected fold-change values in the description of RT-qPCR data (see p. 12, line 324). We believe these changes will improve the readability of the figures and their connection to the text.
- The manuscript requires clearer organization and improved presentation. The introduction is overly long and contains background information not directly relevant to ADAMTS6. The Results section would benefit from a more structured narrative that guides the reader through the findings. Figures would need simplification and reorganization to enhance readability.
Authors: Dear Reviewer #2, Thank you for pointing out the problems in the presentation of our study. We have made substantial changes to the text to improve its presentation. The Introduction section has been shortened and now contains only information directly related to ADAMTS6 (see pp. 1–2, lines 38–56). We have added a structured narrative to the Results sections, including two main points: (1) ADAMTS6 upregulation serves as a marker of EMT in LUAD cells (see p. 3, lines 70–72; p. 4, lines 109–110; p. 6, lines 153–155; p. 6, lines 157–158; p. 9, lines 237–239), and (2) ADAMTS6 plays a positive regulatory function in EMT in LUAD cells (see p. 4, lines 111–113; p. 9, lines 241–242; p.13, lines 352–353). The Discussion section has also been significantly changed to fit the new narrative (see pp. 15–18, lines 396–526). Minor changes have been introduced in the Abstract and Conclusion sections (see p. 1, lines 20–24; p. 22, lines 726–732). Simplification of figures was already mentioned in the response to question 4. Taken together, we believe these changes will help readers navigate our findings and enhance readability.
We hope that corrected version of the manuscript will be acceptable for publication in the International Journal of Molecular Sciences.
Respectfully yours,
Dr. Andrey Markov
Akademgorodok, Russia

Round 2
Reviewer 2 Report
Comments and Suggestions for Authors
The authors have made meaningful and well-justified revisions that improve the manuscript’s scientific rigor, clarity, and logical coherence. While some experimental limitations remain—particularly the absence of rescue or overexpression approaches—the authors have acknowledged these appropriately, and the current dataset supports the main conclusions within the stated limitations.
I find the revised manuscript acceptable for publication after these changes.